# Lenalidomide and Pomalidomide Improve Function and Induce FcγRI/CD64 in Multiple Myeloma Neutrophils

**DOI:** 10.3390/biomedicines9101455

**Published:** 2021-10-13

**Authors:** Alessandra Romano, Nunziatina Laura Parrinello, Marina Parisi, Vittorio Del Fabro, Angelo Curtopelle, Salvatore Leotta, Concetta Conticello, Francesco Di Raimondo

**Affiliations:** 1Dipartimento di Chirurgia e Specialità Medico-Chirurgiche, Università degli Studi di Catania, 95123 Catania, Italy; diraimon@unict.it; 2Division of Hematology, AOU “Policlinico—Vittorio Emanuele”, 95123 Catania, Italy; lauraparrinello@tiscali.it (N.L.P.); marinaparisi@hotmail.it (M.P.); vdelfabro@yahoo.it (V.D.F.); angelocurtopelle@gmail.com (A.C.); leotta3@yahoo.it (S.L.); ettaconticello@gmail.com (C.C.)

**Keywords:** multiple myeloma, neutrophil, phagocytosis, CD64

## Abstract

**Background** Myeloid dysfunction is an emerging hallmark of microenvironment changes occurring in multiple myeloma (MM). Our previous work showed that FcγRI/CD64 overexpression in neutrophils of newly diagnosed MM patients is associated to inferior outcomes, reduced oxidative bursts and phagocytosis, with an increased risk of bacterial infections. Pomalidomide is a novel immune-modulatory drug approved for relapsed/refractory patients (RRMM), with drug-related neutropenia as major limitation to treatment. **Patients and methods** Herein, we describe a prospective analysis of 51 consecutive RRMM patients treated with pomalidomide and dexamethasone (PomDex) from March 2015 through December 2016, associated with secondary prophylaxis with filgrastim (G-CSF) in case of neutrophil count <1500 cells/μL. Neutrophil function was investigated by flow cytometry, including the phagocytosis, oxidative bursts, and median fluorescence intensity of FcγRI-CD64. Controls included a group of newly diagnosed symptomatic MM (NDMM), asymptomatic (smoldering myeloma, MGUS) and healthy subjects referred to our Center in the same time-frame. **Results** Compared to controls, RRMM neutrophils had higher expression of FcγRI/CD64 and lower phagocytic activity and oxidative bursts. We maintained median leukocyte counts higher than 3.5 × 10^9^/L for 6 cycles, and median neutrophil counts higher than 1.5 × 10^9^/L, with only 6 (11%) patients developing grade 3–4 infections, without pomalidomide dose reduction. After 4 cycles of PomDex, FcγRI/CD64 was further increased in neutrophils, and phagocytic activity and oxidative bursts recovered independently from filgrastim exposure and the quality of hematological responses. Similarly, in NDMM patients, lenalidomide but not bortezomib upregulated FcγRI/CD64 expression, improving phagocytic activity and oxidative bursta as tested in vitro. **Conclusions** Our combined biological and clinical data provide new information on the ability of pomalidomide and lenalidomide to modulate the functional activity of neutrophils, despite their chronic activation due to FcγRI/CD64 overexpression.

## 1. Introduction

Multiple myeloma (MM) is the second most frequent hematological malignancies, due to the abnormal proliferation of neoplastic plasma cells in a supportive bone marrow microenvironment (BMM) which alters immune homeostasis, leading to immune-suppression, impaired lymphocyte function and hypogammaglobulinemia, with consequent increased prevalence of infections [1,2,3,4]. In a large, population-based study in Sweden, asymptomatic patients with MGUS had increased risk of developing bacterial (pneumonia, osteomyelitis, septicemia, pyelonephritis, cellulitis, endocarditis, and meningitis), and viral (influenza and herpes zoster) infections at the 5- and 10-year follow-ups [4]. This peculiar clinical feature is associated to early changes occurring in the BMM which are conveyed by deranged interferon signaling, as shown in in vitro and in vivo studies [5,6]. In response to soluble factors released by neoplastic plasma cells, several transcriptional derangements can occur in the mature myeloid compartment, as identified first by our group by the gene expression profiles of immunomagnetically sorted neutrophils [3] and recently confirmed by the Spanish group using RNAseq upon FACSorting of several neutrophil subsets [7]. Indeed, in the MM bone marrow, there is a correlation between the clinical significance, immunosuppressive potential, and transcriptional network of mature neutrophil subsets. For example, T cell proliferation decreased in presence of mature neutrophils [2,3,8], and the cytotoxic potential of T cells engaged by a BCMAxCD3 bispecific antibody increased notably with the depletion of mature neutrophils [7].

Fc receptors (FcRs) control the humoral and innate immunities, which are essential for appropriate responses to infections and the prevention of chronic inflammation or auto-immune diseases. Following their crosslinking by immune complexes, FcRs play various roles such as the modulation of the immune response through released cytokines or phagocytosis. The engagement of FcγRs expressed on neutrophils surface is an emerging piece of the crosstalk between B-cells and neutrophils, which can act as immune effector cells to regulate phagocytosis and inflammatory mediator release [9], or sustain plasma cell differentiation in the niche [10,11]. FcγRI/CD64 promotes the recognition and phagocytosis of IgG coated particles and activates the oxidative responses of neutrophils, and it can be increased in all cases of sterile inflammation in which phagocytosis is impaired [12].

Since our previous work showed an increase in the expression of the high-affinity Fc receptor for IgG (FcγRI, CD64) in MM neutrophils [3], we aimed to investigate its modulation during anti-MM treatment in both newly-diagnosed (NDMM) and refractory (RRMM) settings.

Immunomodulatory agents, including lenalidomide and pomalidomide, are a new category of drugs acting on the immune system [13]. Pomalidomide is an immunomodulator with multiple activities: immunomodulating, antiangiogenetic and antineoplastic. Pomalidomide appears to inhibit TNF-alpha production, enhance the activity of T cells and natural killer (NK) cells, and enhance antibody-dependent cellular cytotoxicity (ADCC) [13,14,15]. Despite these properties, pomalidomide therapy is characterized by a high incidence of neutropenia and febrile neutropenia, which are also responsible for frequent treatment interruptions or dosage reductions that may compromise the efficacy of the treatment [15,16,17,18,19,20]. To establish if neutrophil function is still compromised in RRMM and if pomalidomide or supportive care with filgrastim can affect the expression of FcγRI/CD64 on surface neutrophils, we designed a single-center prospective observational study in double-refractory RRMM patients treated uniformly with PomDex. Our findings were compared with a group of newly diagnosed symptomatic MM and asymptomatic (smoldering myeloma, MGUS) patients referred to our center in the same timeframe.

## 2. Materials and Methods

### 2.1. Patients, Controls, and Samples

A total of 215 peripheral blood EDTA-anticoagulated samples from 20 controls and 115 adult plasma cell disorder patients were studied to evaluate the expression of FcγRI-CD64 on surface neutrophils. All subjects were referred to our outpatient service between March 2015 and December 2016.

The 215 consecutively recruited samples corresponded to: 20 healthy donors, 25 low-risk MGUS (according to IMWG criteria), 10 smoldering MM not requiring treatment, 29 newly diagnosed MM (NDMM) undergoing standard treatment according to local policy, tested two times (before and after induction therapy) and 51 double-refractory MM (RRMM) patients undergoing treatment with the PomDex salvage regimen, tested two times, before and after treatment. All samples were processed within 24 h after collection.

For patients with active MM requiring treatment, samples were collected before and after four 28-day cycles of treatment. Key exclusion criteria were pregnancy, active infection, exposure to G-CSF or high-doses steroids in the last 28 days, chronic inflammatory or immune-mediated diseases (e.g., uncontrolled diabetes, autoimmune diseases treated with immune-modulatory agents or steroids), and acute or chronic viral infections, to avoid any interference on immune-regulatory mechanisms. Thus, none of the recruited patients was receiving medical treatments that could have an impact on their immune condition.

The study was approved by the local ethics committee (Comitato Etico Catania 1, #19351, 04.12.2009, https://www.policlinicovittorioemanuele.it/comitato-etico-catania-1) and written informed consent was given by each participant according to the Declaration of Helsinki.

### 2.2. Patients’ Treatments

For the 29 NDMM patient candidates for autologous stem cell transplantation (ASCT), treatment consisted of bortezomib 1.3 mg/m^2^ and dexamethasone 40 mg weekly on days +1, +4, +8, +11, and thalidomide 100 mg daily per 28-day cycle (VTD) or Lenalidomide 25 mg on day 1 through 21 and dexamethasone 40 mg weekly per 28-day cycle (Rd) for four cycles, followed by stem cell mobilization and collection and further ASCT consolidation.

For the 51 RRMM patients, treatment consisted of 28-day cycles of PomDex regimen with pomalidomide given 4 mg daily per os on days 1–21 of each 28-day cycle, and dexamethasone 40 mg weekly (for <75 years patients) and 20 mg weekly (for ≥75 years patients) until progression.

From July 2013 to January 2016, a total of 51 consecutive RRMM patients with a median age of 63 years (range 43–83) received PomDex. At PomDex initiation (median of 28 months after MM diagnosis), all patients had received bortezomib and lenalidomide (double refractory). The median number of prior therapies was 4 (range 2–7), including 38 double refractory patients (68%). At baseline, 33 patients (59%) had severe anemia, 16 (28%) were neutropenic, 5 (9%) had a low platelet count, 14 (25%) had eGFR <45 mL/min, and 10 had an extra-medullary disease (18%). The baseline characteristics of patients included in the study are summarized in Table 1, as distinguished based on the treatment received and need of filgrastim support in the RRMM setting.

White blood cell count and types (neutrophil, lymphocyte, eosinophil, and monocyte) were determined by the electrical impedance method in anautomatic blood counter device (Beckman Coulter LH 750). Each patient’s medical history was recorded on day 1 of each cycle. Physical examinations were conducted, and blood was collected for hematology, renal and liver function tests on day 1 of each course. Infective events were graded using the National Cancer Institute Common Terminology Criteria for Adverse Events (NCI-CTC) criteria.

### 2.3. Supportive Care

In our center, we adopted a strong supportive program to maintain absolute neutrophil count (ANC) over 1.5 × 10^9^ cells/mmc, in an attempt to reduce incidences of neutropenia and its complications, but at the same time to avoid dosage reductions of pomalidomide. In 15 patients who were neutropenic at the beginning of therapy (median ANC 0.8 × 10^9^/L), the injection of filgrastim (5 mcg/kg/day) for three consecutive days and then once a week was started simultaneously to PomDex. For all other patients, after an ANC evaluation at the beginning of every cycle, filgrastim was administered one week later if ANC was between 1 and 1.5 × 10^9^/L for one or two consecutive days (depending on ANC value and patient frailty). If the ANC value was <1 × 10^9^/L, filgrastim was administered immediately for two consecutive days. Thus, a total of 26/51 (51%) patients received prophylaxis with filgrastim, biosimilar or originator, at the dose of 5 mcg/kg/day for at least 1 day, including 15 primary and 11 secondary prophylaxes, named group B. Patients who did not receive filgrastim support were named group A (N = 25) (Table 1).

For both NDMM and RRMM patients, concomitant medications included agents for thrombo-prophylaxis with low-dose aspirin for low-risk patients and with low-molecular-weight heparin for high-risk patients [15], and anti-infectious prophylaxis, consisting of Trimetoprim and Sulfametoxazole 800 mg twice daily for two days per week with Acyclovir 400 mg daily. In RRMM patients, secondary prophylaxis with sub-cutaneous G-CSF was given if the neutrophil count (ANC) was ≤1.5 × 10^9^/L, to avoid pomalidomide dosage reduction. NDMM patients did not receive any prophylaxis with subcutaneous G-CSF.

All evaluations were performed within the first four cycles to register changes in neutrophil function before and after four cycles of treatment, and number of infectious episodes.

### 2.4. Immunephenotype and Function Evaluation of Neutrophils by Flow Cytometry

All samples in this study were stained following the EuroFlow standard operating procedures for sample preparation and staining for surface membrane markers [21]. Expression levels of individual markers were reported as median fluorescence intensity values (MFI; arbitrary units). To evaluate neutrophil function, we collected fresh samples of peripheral blood in ethylenediaminetetraacetic acid (EDTA) and heparin tubes processed within two hours to avoid artifacts. Fifty μL of EDTA peripheral blood was stained with 10 μL of the following monoclonal antibodies CD45-ECD (clone J33), CD15-PB (clone 80H5), CD64-FITC (clone 22), CD16a-PC7 (clone 3G8) and respective isotypic controls (Beckman Coulter). Viability was assessed by annexin exclusion. After red cells lysis with PharmLyse (BD Biosciences), samples were processed for flow-cytometry using a Navios 10/3 flow-cytometer (Beckman Coulter).

Phagocytic and oxidative burst activities were detected from heparin blood using the Phagotest Kit (Opregen Pharma, Heidelberg, Germany) and the Phagoburst Kit (Glycotope Biotechnology GmbH, Heidelberg, Germany), following the manufacturer’s instructions. Cells were gated through the scatter parameters (forward, FCS versus side scatter, SSC) and their fluorescence histograms were analyzed by an Epics XL-MCL four-color flow cytometer. The results were expressed as percentage of fluorescent cells in the population studied, calculated by subtracting the percentage of the negative control sample (<1%) from the positive sample.

EXPO32 ADC Analysis Software (Beckman Coulter) was used for data analysis. Neutrophils were selected according to forward and 90° light scatter parameters. For each sample, 20,000 events were acquired. Results were expressed as mean fluorescence intensity (MFI) corrected for values of nonspecific binding.

### 2.5. Statistical Analysis

Descriptive statistics were generated for analysis of the results and *p*-values under 0.05 were considered significant. All calculations were performed using Graph Pad Prism version 6.00 for Windows, Graph Pad Software (La Jolla, CA, USA). The Wilcoxon or Friedman tests and the Mann–Whitney U or the Kruskal–Wallis tests were used to assess the (two-sided) statistical significance of the differences observed between ≥2 groups for paired and unpaired variables, respectively. For the correlation studies, the (two-sided) Spearman’s rho (ρ) test for non-parametric paired data was employed.

For each measurement, the median and IQR for the non-symmetric variables were calculated and compared among healthy MGUS and MM subjects using a two-tailed Student’s t test with Bonferroni correction.

Spearman’s correlation test was applied to measure the monotonic relationship between the two continuous random variables CD64 and phagocytic activity of neutrophils.

All samples evaluated were blindly analyzed during the experimental phase.

## 3. Results

### 3.1. Neutrophil Dysfunction across Plasma Cell Dyscrasias

The neutrophil population in peripheral blood can be distinguished by flow cytometry based on forward- and side-scatter characteristics. We included, as additional markers for viability and lineage identification, fluorescently labelled antibodies against CD15 and CD45.

As summarized in Figure 1A, median fluorescence intensity (MFI) of FcγRI/CD64 was progressively increased from MGUS through active MM, with the highest value being in the RRMM setting, and no differences between MGUS and healthy subjects. However, FcγRI-CD64 expressions were not correlated, either to the amount of the monoclonal component, or the infiltration of plasma cells in the bone marrow or the absolute neutrophil count (data not shown).

Since the literature reported that increased expression of FcγRI-CD64 is associated to neutrophil activation and phagocytosis priming, and our previous work showed an impairment of phagocytosis and oxidative bursts in NDMM, we investigated the neutrophil function in the same samples, taking advantage of ex-vivo measurements of phagocytic activity and oxidative bursts.

As shown in Figure 1B, phagocytic activity was progressively reduced among plasma cell neoplasms (*p* < 0.001, ANOVA-test), with the lowest amounts in NDMM (mean activity 48.4 ± 12.1%) and RRMM (mean activity 31.5 ± 8.7%, *t*-test, *p* < 0.001), and no differences between MGUS and sMM (respectively, mean activity 72.1 ± 5.6% versus 73.0 ± 5.4%, *t*-test, *p* = 0.92).

Oxidative burst activity was impaired among plasma cell neoplasms tested (*p* < 0.001, ANOVA-test), with the lowest amounts in NDMM (mean activity 71.2 ± 13.3%) and RRMM (mean activity 60.0 ± 15.1%, *t*-test, *p* < 0.001), and significant differences also between MGUS and sMM (respectively, mean activity 78.8 ± 8.1% versus 88.1 ± 7.4%, *t*-test, *p* < 0.001), as shown in Figure 1C.

### 3.2. Lenalidomide and Pomalidomide Can Increase Expression of FcγRI/CD64

Treatment with IMiDs further increased the expression of FcγRI/CD64, as seen in patients evaluated after 4 Rd cycles (*p* < 0.0001) or 4 PomDex cycles (*p* < 0.0001), while 4 cycles of bortezomib-based treatment did not affect the FcγRI-CD64 (*p* = 0.21, Table 2).

In the RRMM setting, we maintained for 6 cycles median leukocyte counts higher than 3.5 × 10^9^/L and median neutrophil counts higher than > 1.5 × 10^9^/L, with the same trend in both cohorts, despite the median values in group B being significantly lower than in group A (ANOVA test, *p* = 0.02). There were no significant differences among cohort A and B, except the reduced medullary reserve due to the increased plasma cell infiltration rate in cohort B and consequent lower blood counts (Table 1). In group A, 25 patients received a total of 147 cycles of PomDex, and no G3/G4 neutropenia or clinically relevant infections were recorded. In group B, 18/26 (69%) patients developed G3-G4 neutropenia; 15 patients treated with primary filgrastim prophylaxis received 70 cycles of PomDex treatment, and were complicated by 19 episodes of G3 neutropenia and no cases of G4; while 11 patients treated with secondary filgrastim prophylaxis for a total of 61 cycles had 13 G3/G4 episodes of neutropenia. Only 5/26 (19%) patients experienced grade 3–4 infections, mostly pneumonia that was resolved with antibiotic treatment, not requiring hospitalization.

The upregulation of FcγRI/CD64 was independent from G-CSF exposure or the depth of the achieved response. Indeed, FcγRI/CD64 further increased upon PomDex treatment in both cohorts (MFI 19.9 ± 0.6 versus 32.9 ± 0.9 in cohort A, *p* < 0.0001; MFI 27.2 ± 0.6 versus 32.1 ± 0.9 in cohort B, *p* < 0.0001).

### 3.3. Lenalidomide and Pomalidomide Can Improve Phagocytic Activity in Neutrophils

In NDMM patients, median phagocytic activity increased after 4 cycles of lenalidomide-dexamethasone (median phagocytic activity 46.0 vs. 54.5%, *p* = 0.007), but not after 4 cycles of bortezomib-thalidomide-dexamethasone (median phagocytic activity 44.0 vs. 45.0%, *p* = 0.86, Table 3).

In RRMM, four cycles of pomalidomide-dexamethasone significantly increased the ex-vivo phagocytic function (median phagocytic activity 40.1 vs. 69.6%, *p* < 0.0001, Table 3). 

The improvement in neutrophil function was not due to G-CSF exposure or the depth of the achieved response (data not shown). Indeed, phagocytic activity was fully recovered after 4 cycles of PomDex in both cohorts (35.7 ± 2.8 versus 62.2 ± 2.1%, *p* < 0.0001 in cohort A; 42.9 ± 2.9 versus 74.3 ± 1.7%, *p* < 0.0001 in cohort B). 

After first-line treatment, FcγRI-CD64 expression and phagocytosis quantification were correlated in neutrophils obtained from patients treated with lenalidomide (r-square 0.38, *p* = 0.02, Figure 2A), but not with bortezomib (r-square 0.06, *p* = 0.78, Figure 2B).

In RRMM patients, after four cycles of pomalidomide-dexamethasone, FcγRI-CD64 expression and phagocytosis quantifications were correlated (r-square 0.64, *p* = 0.0004, Figure 2C), independently from G-CSF exposure (data not shown). In RRMM, four cycles of pomalidomide-dexamethasone significantly increased the oxidative burst activity (median activity 40.1 vs. 69.6%, *p* < 0.0001). Oxidative bursts fully recovered after 4 cycles of treatment independently from G-CSF exposure (86.1 ± 3.1 versus 63.9 ± 3.3%, *p* < 0.0001 in group A; 89.6 ± 2.3 versus 58.2 ± 3.7%, *p* < 0.0001 in group B).

## 4. Discussion

Pomalidomide was approved in 2015 in Italy for RRMM patients in combination with dexamethasone, thanks to its favorable toxicity profile and efficacy [14,15,16,17,18,19,20]. Herein, we evaluated the neutrophils function in both NDMM and RRMM patients based on immune-phenotype and functional assays before and after treatment with the immunomodulatory drugs lenalidomide and pomalidomide, compared to healthy subjects or NDMM subjects treated with bortezomib-based regimens.

The first relevant finding of our study is that FcγRI/CD64, a marker of neutrophil activation, increases in RRMM and further increases after PomDex. Human FcγRI/CD64 is a 72-kDa transmembrane glycoprotein that recruits monomeric IgG1, IgG3 and IgG4, but not IgG2 [22]. Under physiological conditions, G-CSF upregulates FcγRI/CD64 on neutrophil surfaces and promotes tumor cell killing in vitro [23]. The expression level of FcγRI/CD64 is also inducible on neutrophils in response to TNF-α, interferon-γ, interleukin-12 or G-CSF [24] and its expression on neutrophil surfaces can resemble the microenvironment dysfunction occurring in RRMM patients. In several hematological settings, neutrophils are primed and show phenotypic alterations, such as the elevation of the FcγRI/CD64 and CD54 markers, which correlate with disease activity [25]. Similar to chronic inflammatory diseases, including primary vasculitis, pancreatitis, and tuberculosis [26,27], the upregulation of FcγRI/CD64 could mirror the extended life span of neutrophils and their acquisition of immune-suppressive activity, contributing to T-cell dysfunction, immune escape and neoplastic cell expansion [28]. In our series, changes in neutrophil functions occurred early during the treatment and were not associated with G-CSF administration and the depth of achieved response as part of the consequences of pomalidomide on the network of immune cells in a context of sterile inflammation [1].

The second clinical implication of our work is that pomalidomide-related neutropenia was not associated with an increased risk of severe infection. Neutrophils could recover their phagocytic activity after exposure to lenalidomide and pomalidomide, but not bortezomib.

In tumor-bearing subjects, febrile neutropenia is one of the most relevant chemotherapy-related adverse events [29], leading to morbidity, mortality, and high costs of patient management [29,30]. The use of G-CSF is recommended by the American Society of Clinical Oncology and the National Comprehensive Cancer Network in patients receiving cytotoxic agent treatments to reduce the risk of neutropenia and its complications, enabling safe and effective chemotherapy dose intensity [31]. Primary prophylaxis is indicated when the overall FN risk is greater than 20%. In the other cases (≤20% of risk) secondary prophylaxis is recommended for patients who experienced a neutropenic complication during a previous cycle of chemotherapy and consists of post- chemotherapy G-CSF administration if ANC is <1 × 10^9^/L. Based on current guidelines, in RR-MM patients experiencing grade 4 neutropenia or febrile neutropenia, pomalidomide should be withheld and re-started at a lower dosage when absolute neutrophil counts have recovered to ≥500 cells/μL [29,30,31].

In our series, PomDex was well tolerated, with only 6 cases of active infection (mainly pneumonia), not requiring hospitalization. These real-life findings are in line with those reported by phase II–III clinical trials which investigated the safety and tolerability of pomalidomide in RRMM patients.

When compared to lenalidomide, pomalidomide induced higher rates of febrile neutropenia [30]. In MM002, MM003 and MM010 trials, the most frequent hematological adverse event of grade 3/4 was neutropenia (between 34% and 55%) with grade 3–4 infections ranging from 14 to 44%, with consequent pomalidomide dose reductions and interruptions occurring in 24.2% and 66% of patients, respectively [29]. In the MM-002 trial, dose reduction occurred in 29% of patients, with 51% of temporary interruptions [29]. In the MM-010 trial, 22% of patients in the PomDex arm needed a dose reduction, with a 66% drop-off for adverse events [20,29]. In the MM-03 trial, G-CSF was used in 47% of patients, with low incidences of febrile neutropenia and most grade 3–4 infections being unrelated to neutropenia [22,32]; pomalidomide dosage was reduced in 27% of patients, and interrupted in 67% of cases for adverse events and low compliance [32]. Similar results could also be achieved in real-life experiences [14,33,34,35].

Based on our policy, filgrastim was given one week after starting pomalidomide therapy when neutrophils were ≤1.5 × 10^9^/L in attempt to reduce the incidence of neutropenia and its complications. The prediction and management of adverse events is essential for the safe use of pomalidomide and to improve clinical outcome [16,33], as recently suggested also by other small single-center experiences. One strategy could include injection of peg-filgrastim once monthly [35] or administration of G-CSF 5 mcg/kg on days 22–28 every 28 days to maintain the 5 mg full dose of pomalidomide [36]. Indeed, giving pomalidomide 5 mg and G-CSF support, the ORR was 75%, median PFS 17.5 months and the OS was 100% at a median follow up of 35 months [36]. The addition of G-CSF by increasing the neutrophil count may further reduce the incidence of infections, thus allowing us to avoid stopping or delaying treatment due to toxicity, but the recovery of phagocytosis and oxidative bursts in RRMM neutrophils in patients treated with a PomDex regimen is not a consequence of G-CSF exposure.

## 5. Conclusions

Our combined biological and clinical data provide new information on the ability of pomalidomide to modulate the functional activity of neutrophils, despite their chronic activation due to CD64 overexpression. Pomalidomide improves the neutrophil functional profile, increasing phagocytic activity and oxidative bursts.

## Figures and Tables

**Figure 1 biomedicines-09-01455-f001:**
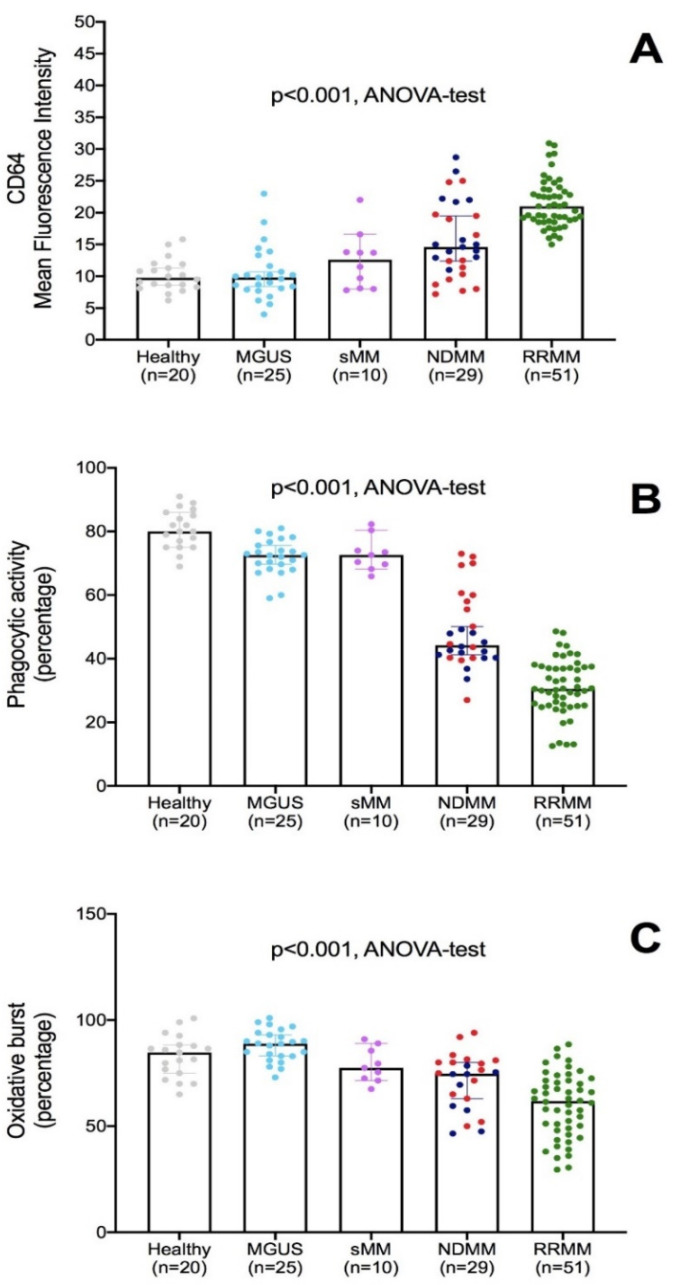
Functional evaluation of neutrophils across plasma cell dyscrasias. (**A**) Mean fluorescence intensity for FcγRI/CD64 expressed on neutrophils, across healthy, MGUS, smoldering myeloma, newly diagnosed myeloma (NDMM) and double refractory myeloma (RRMM). After exposure to E-coli bacteria opsonized with IgG and a complement of pooled sera, percentages of phagocytic activity (**B**) and oxidative bursts (**C**) were detected by flow cytometry. For more robust statistical evaluation, MFI values were converted to a resolution metric, such as the RD defined as (Mediantreatment − Mediancontrol)/(rSDtreatment  +  rSDcontrol) to further perform a *t*-test to compare results of different experiments and runs. Bars represent medians and rSD, and each dot represents the value of a single patient. Colors represent different therapeutic approaches and disease categories (grey: healthy subjects; clear blue: MGUS; purple: sMM; red: newly diagnosed patient candidates to bortezomib-based regimens; dark blue: newly diagnosed patient candidates to lenalidomide-based regimens; green: RRMM patient candidates to Pom-Dex).

**Figure 2 biomedicines-09-01455-f002:**
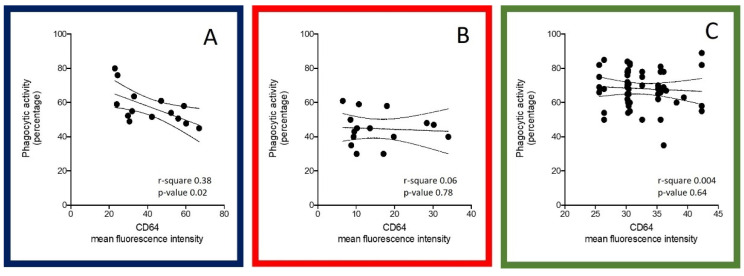
Correlation between phagocytosis recovery and expression of FcγRI/CD64 after treatment in NDMM and RRMM patients. After exposure to *E. coli* bacteria opsonized with IgG and a complement of pooled sera, percentages of phagocytic activity were detected by flow cytometry in controlled conditions in newly diagnosed multiple myeloma patients before and after lenalidomide-based induction (**A**), and before and after bortezomib-based induction (**B**). Percentage of phagocytic activity in double refractory multiple myeloma patients before and after PomDex (**C**). For more robust statistical evaluation, percentage values were converted to a resolution metric, such as the RD, defined as (Mediantreatment − Mediancontrol)/(rSDtreatment  +  rSDcontrol), to further perform *t*-tests for paired samples to compare results of different experiments and runs. Spearman’s correlation was applied to measure monotonic relationships between the two continuous random variables CD64 and the phagocytic activity of neutrophils.

**Table 1 biomedicines-09-01455-t001:** Baseline characteristics of patients included in the study.

	NDMM	NDMM	RRMM Group A	RRMM Group B
	(Rd)	(VTD)		
	N = 14	N = 15	N = 25	N = 26
Median age, years (range)	68 (43–73)	63 (50–66)	61 (43–83)	63 (50–75)
Gender, Males/Females, n (%)	8/6 (60/40)	7/8 (47/53)	15/10 (60/40)	15/11 (60/40)
**Isotype**				
IgG, n (%)	9 (64)	8 (53)	16 (64)	21 (81)
IgA, n (%)	5 (28)	4 (27)	7 (28)	2 (8)
Light-chain only, n (%)	0 (0)	3 (20)	2 (8)	3 (12)
**ISS stage at baseline**				
I, n (%)	7 (50)	3 (20)	7 (28)	7 (27)
II, n (%)	4 (29)	4 (27)	9 (36)	6 (23)
III, n (%)	3 (21)	8 (53)	9 (36)	13 (50)
**Risk assessment at baseline**				
FISH not available/failed, n (%)	2 (14)	3 (20)	14 (56)	21 (81)
High risk cytogenetic aberrations, n (%)	3 (21)	4 (27)	9 (36)	7 (27)
Standard risk cytogenetic aberrations, n (%)	9 (64)	8 (53)	2 (8)	1 (4)
Median LDH, UI (range)	233 (127–690)	241 (139–650)	247 (127–691)	267 (170–637)
Median β2-M, mg/L (range)	4.9 (1.3–15.7)	5.7 (1.8–18.9)	3.3 (1.3–15.7)	4.3 (1.8–18.9)
Median PC infiltration, % (range)	80 (60–90)	70 (50–90)	40 (40–90)	60 (30–90)
**Blood counts at baseline**				
Median Hb, g/L (range)	10.1 (5.6–10.5)	9.9 (6.8–10.3)	10.3 (7.6–13.5)	9.6 (7.2–11.3)
Median WBC, ×10^3^ cells/mL (range)	4.1 (1.9–6.6)	3.9 (2.1–7.3)	3.6 (1.9–6.1)	3.6 (2.5–7.5)
Median ANC, ×10^3^ cells/mL (range)	2.3 (1.1–5.1)	2.1 (1.5–6.4)	1.8 (0.9–5.1)	1.1 (0.4–6.4)

Abbreviations: LDH: lactate dehydrogenase, β2-M, beta-2 microglobulin PC, plasma cells, Hb, hemoglobin; WBC, white blood cells; ANC, absolute neutrophil count; Rd, lenalidomide and dexamethasone; VTD, bortezomib, thalidomide and dexamethasone; NDMM, newly diagnosed multiple myeloma; RRMM, double refractory multiple myeloma.

**Table 2 biomedicines-09-01455-t002:** Median fluorescence intensity (MFI) of FcγRI/CD64 across MM patients included in the study, before and after 4 cycles of anti-MM treatment.

	NDMM (N = 14)			NDMM (N = 15)			RRMM (N = 51)		
	Baseline	After 4 Cycles Rd	*p*-Value *	Baseline	After 4 Cycles VTD	*p*-Value *	Baseline	After 4 Cycles PomDex	*p*-Value *
Median CD64	15.0	37.7	** *<0.0001* **	12.4	10.7	0.13	21.1	32.6	** *<0.0001* **
25% Percentile	13.7	28.5		8.7	9.3		18.7	30.2	
75% Percentile	22.0	56.8		19.5	19.8		23.6	37.2	
95% CI of median	13.0–22.2	24.4–59.1		8.7–19.5	9.3–19.8		19.3–22.5	30.4–35.6	

* Wilcoxon test. Abbreviations: sMM, smoldering multiple myeloma; NDMM, newly diagnosed multiple myeloma; RRMM, double refractory multiple myeloma.

**Table 3 biomedicines-09-01455-t003:** Percentage of phagocytic activity before and after treatment with lenalidomide-, bortezomib- or pomalidomide-based regimens.

	NDMM (N = 14)			NDMM (N = 15)			RRMM (N = 51)		
	Baseline	After 4 Cycles Rd	*p*-Value *	Baseline	After 4 Cycles VTD	*p*-Value *	Baseline	After 4 Cycles PomDex	*p*-Value *
Median Phagoytic activity	46.0	54.5	** *0.007* **	44.0	45.0	0.86	40.1	69.6	** *<0.0001* **
25% Percentile	38.7	50.2		35.6	40.0		27.7	60.1	
75% Percentile	49.0	61.6		54.0	50.0		56.8	78.2	
95% CI of median	38.0–52.0	49.0–63.6		35.6–54.0	40.0–50.0		37.3–44.5	65.0–72.3	

* Wilcoxon test. Abbreviations: NDMM, newly diagnosed multiple myeloma; RRMM, double refractory multiple myeloma; Rd lenalidomide-dexamethasone; VTD: bortezomib, thalidomide, dexamethasone; PomDex: pomalidomide, dexamethasone.

## Data Availability

Raw data are available upon request to the corresponding author.

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
