# Peer review of "Lenalidomide and Pomalidomide Improve Function and Induce FcγRI/CD64 in Multiple Myeloma Neutrophils"

_biomedicines, 2021, doi:10.3390/biomedicines9101455_

Round 1

Reviewer 1 Report

In this paper Dr. Romano et al. show their ability to evaluate the combination between biological and clinical activity in MM pathogenesis, modulating neutrophils function. 

The paper is well written and data are concordant with recent literature papers.

Minor Remarks:

In the text all legends to the figures are very poor and not detailed. Authors should improve them by specifing the details of the figures described for each legend.

Author Response

Thanks a lot for your suggestions. Please find in attachment the authors' rebuttal.

Reviewer 2 Report

This manuscript reports on dysfunctions of neutrophils in patients with refractory/relapsing multiple myeloma and on the efficacy of growth factors prescribed concomitantly to classical myeloma therapy in improving neutrophil functions and limiting infections.

This works follows a previous study of this group showing the spontaneous activation of neutrophils in myeloma patients characterized by overexpression of CD64. Here lenalidomide but not bortezomib induced an increased expression of CD64 on neutrophils, together with improved neutrophil functions after lenalidomide, yet with a negative correlation. The authors do not explain this phenomenon. They also report on the effect of pomalidomide on increased CD64 expression and neutrophil activation, with or without growth factor, indicating the absence of benefit of this addition.

The title could better indicate that the study focused on imids, i.e. “IMIds improve function and inducs FcγRI/CD64 in multiple myeloma neutrophils”. Otherwise, since the main message is that of low infections rate with pomalidomide, one could consider :” Pomalidomide improves neutrophils function and prevents infection in multiple myeloma”.

Although these data are of interest, their presentation is very confusing and should be clarified. Moreover, the discussion is overlong and digresses from what was effectively studied and analyzed. Finally, the conclusion is not supported by the data shown.

Indications are provided below.

Page 2, line 16 : the sentence is awkward, explain about the CD11b CD16

Pages 2/3 and table 1 : check numbers, the total as shown does not amount to 215

Page 4 replace bis in die by twice daily

Page 4: what do the authors mean by “for surface membrane markers alone and surface markers”?

Page 5: what do the authors mean by “green fluorescence”?

Figure 1 and table 2 are redundant.

Page 7 line 29 : “but not bortezomib”

Page 7 : please show oxidative burst data

Pages 7-8 : results of oxidative burst and CD64 expression are presented in a confused manner. Please consider rewriting, focusing consecutively on each condition.

Page 8 : the whole story of filigrastim addition to therapy should be moved to the treatment paragraph and, as mentioned just above, the effects of treatement on CD64 expression and neutrophil functions should be reported separately for each regimen for the sake of clarity.

Page 10: the legend of Figure 3 should follow the order of the graphs. Moreover, it would be more (or at least as) interesting to show the absence of difference related to filigrastim supplementation rather than repeating data already shown for the whole pomalidomide group.

Page 11: the discussion should focus on the study presented. Elaborating on the effect of CD64 expression and monoclonal antibody therapy is out of scope of this work since none of these patients received such immunotherapy. This overlong hypothesis should be shortened as a small conclusive hypothesis that should be further documented. The same goes for the hypotheses raised about T-cell functions and cytokine production.

Page 13 : the reference list is over long and will be shortened by focusing on the study at hand. Please also remove caps in titles including them for each word (Pubmed no longer provides this service of homogenization).

Overall, please check the whole manuscript for English grammar.

Author Response

(The authors gave the same response as above.)
